# Robust Invariant Representation Learning by Distribution Extrapolation

## Abstract

Invariant risk minimization (IRM) aims to enable out-of-distribution (OOD) generalization in deep learning by learning invariant representations. As IRM poses an inherently challenging bi-level optimization problem, most existing approaches—including IRMv1—adopt penalty-based single-level approximations. However, empirical studies consistently show that these methods often fail to outperform well-tuned empirical risk minimization (ERM), highlighting the need for more robust IRM implementations. This work theoretically identifies a key limitation common to many IRM variants: their penalty terms are highly sensitive to limited environment diversity and over-parameterization, resulting in performance degradation. To address this issue, a novel extrapolation-based framework is proposed that enhances environmental diversity by augmenting the IRM penalty through synthetic distributional shifts. Extensive experiments—ranging from synthetic setups to realistic, over-parameterized scenarios—demonstrate that the proposed method consistently outperforms state-of-the-art IRM variants, validating its effectiveness and robustness.

## 1 Introduction

In modern machine learning applications, the assumption that training and test data are independent and identically distributed (i.i.d.) often fails to hold, with distribution shifts commonly observed (Quiñonero-Candela et al., 2022). In such scenarios, models are required to generalize to previously unseen data distributions—a challenge known as out-of-distribution (OOD) generalization. Both theoretical analyses and empirical studies have shown that algorithms built on the i.i.d. assumption, and usually realized via the classical framework of empirical risk minimization (ERM) (Vapnik, 1991), tend to perform poorly under distribution shifts (Arjovsky, 2020).

In the context of OOD generalization, invariant risk minimization (IRM) (Arjovsky et al., 2020) has attracted significant attention. IRM is grounded in invariant representation learning, which aims to learn only those features from the input data that are consistently useful for inference under distribution shifts (referred to as invariant features), while discarding all other non-invariant features (referred to as spurious ones). However, IRM models the learning process as a composition of a feature extractor and a classifier, leading to a challenging bi-level optimization problem. As a result, strict implementations of IRM have been found to be practically infeasible (Arjovsky et al., 2020). To address this limitation, (Arjovsky et al., 2020) proposed an approximation called IRMv1, which reformulates the original bi-level problem as a more tractable single-level optimization task. Despite numerous IRMv1 variants (Lin et al., 2022; Ahuja et al., 2022; Zhou et al., 2022; Chen et al., 2022; Zhang et al., 2023), empirical studies have shown that these approaches often fail to outperform a well-tuned ERM-based strategy (Gulrajani & Lopez-Paz, 2020).

This paper argues that the formulation of IRMv1—widely adopted as the foundation for many recent IRM variants—leaves room for improvement. Specifically, the loss or penalty function in IRMv1's single-level optimization task suffers from two key limitations: **(i)** it is highly dependent on the diversity of training environments and does not reliably ensure optimality of the original IRM's lower-level objective; and **(ii)** in over-parameterized settings, this dependence is further amplified, increasing the risk of overfitting to the training distribution.

To address the aforementioned limitations, and drawing inspiration from risk extrapolation (REx) (Krueger et al., 2021), this paper proposes applying distributional extrapolation to the IRMv1 penalty. Since the original IRMv1 penalty function is inherently non-linear with respect to the data distribution, it is not directly compatible with distributional extrapolation. To resolve this, the penalty is reformulated to support extrapolation, ultimately resulting in two novel penalty terms for IRMv1.

Extensive numerical tests—spanning from small-scale settings based on structural equation models (SEMs) (Armitage, 2005) to more practical scenarios involving diverse computer-vision datasets—demonstrate that the proposed method consistently outperforms state-of-the-art IRM variants in terms of both accuracy and various calibration metrics. The implementation of the proposed approach is available at https://anonymous.4open.science/r/IRM_Extrapolation-49E6.

To summarize, the key contributions of this work are as follows.

  **(i)** It is theoretically shown that limited diversity in training environments contributes to the suboptimality of the IRMv1 penalty; see Section 3.

 **(ii)** To enhance environment diversity, distributional extrapolation is incorporated into IRMv1, leading to two novel penalty terms; see Section 4.

**(iii)** The effectiveness of the proposed penalty terms is validated through experiments on synthetic SEMs and more realistic, over-parameterized settings using four computer-vision datasets, demonstrating superior performance over existing state-of-the-art IRM variants in terms of accuracy and calibration metrics; see Section 5.

 **(iv)** It is further shown that the proposed penalty terms can also serve as effective enhancements when integrated into existing IRM variants; see Section 5.

## 2 Background

### 2.1 Notations

The model to be trained takes the form of a function $f\colon \mathcal{X} \to \mathcal{Y}$, where the input space $\mathcal{X} \subset \mathbb{R}^d$ represents the set of input data with dimension $d$, and the output space $\mathcal{Y} \subset \mathbb{R}$ represents the corresponding set of ground-truth labels. Data from environment $e \in E$ are assumed to be generated according to a probability distribution function (PDF) $P_e(x, y)$, where $E$ denotes the set of all possible environments. For a user-defined loss $\ell\colon \mathcal{Y} \times \mathcal{Y} \to \mathbb{R}$, for example, $\ell(y', y) := (y' - y)^2$, $\forall (y', y) \in \mathcal{Y} \times \mathcal{Y}$, *risk* $R_e(f)$ validates model $f$ according to the following definition:

$$R_e(f) := \mathbb{E}_{(x,y) \sim P_e} \{\ell(f(x), y)\}, \tag{1}$$

where $\mathbb{E}\{\cdot\}$ stands for expectation. In the following discussion, it is assumed that $R_e(f) \geq 0$, $\forall f$.

### 2.2 Out-of-distribution generalization

In machine learning, it is commonly assumed that training and test data are i.i.d. To enhance model generalization under this assumption, a variety of techniques have been proposed, including weight decay (Krogh & Hertz, 1991), early stopping (Prechelt, 2002), data augmentation (Perez & Wang, 2017), and ensemble learning (Ganaie et al., 2022). Within the i.i.d. framework, empirical risk minimization (ERM) is widely regarded as one of the most effective approaches for improving generalization. ERM seeks a model which solves the following task:

$$\min_{f\colon \mathcal{X} \to \mathcal{Y}} \sum_{e \in E_{\text{train}}} R_e(f). \tag{ERM}$$

In the context of deep learning, however, the i.i.d. assumption is often violated, and distribution shifts commonly occur in practice (Quiñonero-Candela et al., 2022). That is, the training and test data are drawn from different PDFs. Under such conditions, strategies like ERM—which perform well under the i.i.d. assumption—are known to fail. The task of ensuring model generalization in the presence of distribution shifts is known as out-of-distribution (OOD) generalization, a challenge that has gained significant

attention in recent years. OOD generalization is typically formulated as the following min-max optimization problem (Arjovsky, 2020):

$$\min_{f\colon \mathcal{X}\to\mathcal{Y}} \left( R_{\text{OOD}}(f) \coloneqq \max_{e\in E_{\text{all}}} R_e(f) \right), \tag{2}$$

where $R_{\text{OOD}}$ denotes the worst-case loss across all possible environments, including those generating the test data. In practice, however, it is rarely feasible to account for all possible environments during training. As a result, leveraging the limited set of available training environments effectively to minimize $R_{\text{OOD}}$ becomes a critical objective.

## 2.3 Invariant learning

A central objective in addressing OOD generalization is to ensure that models make consistent predictions across different environments. This consistency is quantified by the environmental invariance of model predictions, defined as (Arjovsky, 2020):

$$\mathbb{E}_{(x,y)\sim P_e}\{y \mid f(x)\} = \mathbb{E}_{(x,y)\sim P_{e'}}\{y \mid f(x)\}, \qquad \forall e, e' \in E_{\text{all}}, \tag{3}$$

that is, the conditional expectation of the label given the model's output remains invariant across all environments. Intuitively, this implies that the relationship between the model's predictions and the true labels is preserved regardless of changes in the data distribution.

However, it is important to note that the condition in (3) does not necessarily guarantee high predictive performance across environments. For instance, a model that produces random predictions would, by definition, maintain the same random predictive behavior in every environment and thus satisfy the environmental invariance condition—despite failing to achieve meaningful accuracy. Furthermore, the well-known lack of environmental invariance in ERM can be attributed to its tendency to rely not only on invariant features but also on spurious ones (Arjovsky et al., 2020; Sagawa et al., 2020).

## 2.4 Invariant risk minimization

One approach to achieving environmental invariance is through representation learning, where models are trained to capture only invariant features (Arjovsky et al., 2020; Ganin et al., 2016). In this context, let the invariant feature extractor be defined as $\Phi\colon \mathcal{X} \to \mathcal{H}$, where $\mathcal{H}$ is a user-defined feature space, and let the predictor (e.g., a classifier) be $\pi\colon \mathcal{H} \to \mathcal{Y}$. The overall model is then given by the composition $f = \pi \circ \Phi$. Within this framework, the condition in (3) can be replaced by the following criterion:

$$\mathbb{E}_{(x,y)\sim P_e}\{y \mid \Phi(x)\} = \mathbb{E}_{(x,y)\sim P_{e'}}\{y \mid \Phi(x)\}, \qquad \forall e, e' \in E_{\text{all}}. \tag{4}$$

The key distinction from (3) is that, in (4), invariance is enforced with respect to the invariant-feature extractor $\Phi$ rather than the full model $f$ which depends also on the predictor $\pi$.

Aiming at (4), invariant risk minimization (IRM) (Arjovsky et al., 2020) has been introduced as the following bi-level optimization task:

$$\min_{\substack{\Phi\colon \mathcal{X}\to\mathcal{H} \\ \pi\colon \mathcal{H}\to\mathcal{Y}}} \quad \sum_{e\in E_{\text{train}}} R_e(\pi \circ \Phi)$$

$$\text{subject to} \quad \pi \in \arg\min_{\bar{\pi}\colon \mathcal{H}\to\mathcal{Y}} R_e(\bar{\pi} \circ \Phi), \forall e \in E_{\text{train}}. \tag{IRM}$$

More specifically, the upper-level problem aims to minimize the sum of losses $R_e$ across all training environments, similar to ERM. However, unlike ERM, the lower-level problem constrains the candidate predictor $\pi$ to those that minimize $R_e$ simultaneously across all training environments. In essence, (IRM) targets feature extractors $\Phi$ that facilitate predictors $\pi$ capable of achieving low loss $R_e$ across all environments, thereby promoting invariant representation learning through $\Phi$, in line with the criterion (4). It is worth noting that the lower-level problem is not a conventional minimization one over an aggregated loss, as is typical in ERM. Instead, it requires minimizing the loss concurrently across all training environments. This requirement makes the exact implementation of IRM particularly challenging, motivating the development of various approximation techniques to make the problem computationally tractable.

## 2.5 IRMv1

IRMv1 (Arjovsky et al., 2020) is one of the most well-known approximations of IRM. In IRMv1, the feature space is set as $\mathcal{H} = \mathbb{R}$ in $\Phi\colon \mathcal{X} \to \mathcal{H}$, and the predictor function $\pi\colon \mathbb{R} \to \mathcal{Y}$ is assumed to be linear. By abuse of notation, this linear predictor function will henceforth be represented by its slope $\pi \in \mathbb{R}$. In this context, $f = \pi \circ \Phi$ is reduced to the simple product form $\pi \cdot \Phi$. Since $R_e(\pi \cdot \Phi) = R_e((\pi/c) \cdot (c\,\Phi))$ for any nonzero scalar $c$, the problem of identifying $(\pi, \Phi)$ in (IRM) exhibits a scaling ambiguity, that is, if $(\pi_{\text{opt}}, \Phi_{\text{opt}})$ is a solution of (IRM), then $\{(\pi_{\text{opt}}/c, c\,\Phi_{\text{opt}}) \mid c \in \mathbb{R} \setminus \{0\}\}$ also solve (IRM). Thus, whenever $\pi_{\text{opt}} \neq 0$, $(1, \pi_{\text{opt}} \cdot \Phi_{\text{opt}})$ solves (IRM). Moreover, when the loss function $R_e(\cdot)$ is convex, loss $R_e(\pi \cdot \Phi)$ is convex in the scalar $\pi$ for any fixed $\Phi$. As a result, under the convexity and differentiability of $R_e$, the lower-level optimization in (IRM) is equivalent to ensuring that the gradient with respect to $\pi$ vanishes at the minimizer $\pi_{\text{opt}}$: $|\nabla_{\pi|\pi=\pi_{\text{opt}}} R_e(\pi \cdot \Phi)|^2 = 0$. Motivated by the aforementioned scaling ambiguity and equivalence, IRMv1 is formulated as the following single-level approximation of IRM (Arjovsky et al., 2020):

$$\min_{\Phi\colon \mathcal{X} \to \mathcal{Y}} \sum_{e \in E_{\text{train}}} \left( R_e(\Phi) + \lambda \, |\nabla_{\pi|\pi=1} R_e(\pi \cdot \Phi)|^2 \right), \tag{IRMv1}$$

where $\lambda \in \mathbb{R}_{++}$—$\mathbb{R}_{++}$ stands for all positive real numbers—is a regularization hyperparameter and $R_e(\cdot)$, in general, is not required to be convex but only differentiable.

IRMv1 has shown promise for out-of-distribution generalization under distribution shifts (Arjovsky et al., 2020), and numerous subsequent methods have been developed by building upon its penalty formulation (Lin et al., 2022; Ahuja et al., 2022; Zhou et al., 2022; Chen et al., 2022; Zhang et al., 2023). Additional details on these variants are provided in Section 6. Nevertheless, empirical evidence suggests that these methods frequently fail to outperform a well-tuned ERM baseline (Gulrajani & Lopez-Paz, 2020).

## 3 The insufficiency of IRMv1 for invariance guarantees

For some $\epsilon \in \mathbb{R}_{++}$, the following relaxation of (IRMv1) facilitates the discussion surrounding Theorem 3.1, which brings forth the insufficiency of IRMv1 to identify those feature extractors $\Phi$ that suppress spurious features and promote invariant ones:

$$\min_{\substack{\Phi\colon \mathcal{X} \to \mathcal{H} \\ \pi\colon \mathcal{H} \to \mathcal{Y}}} \quad \sum_{e \in E_{\text{train}}} R_e(\pi \cdot \Phi)$$

$$\text{subject to} \quad |\nabla_\pi R_e(\pi \cdot \Phi)|^2 \leq \epsilon, \forall e \in E_{\text{train}}. \tag{5}$$

**Assumption 3.1.** *In the context of IRMv1 ($\pi$ is considered to be a scalar) and for any $\Phi$, there exists an $L_\Phi \in \mathbb{R}_{++}$ such that the partial differential operator $\nabla_\pi R_e(\pi \cdot \Phi)$ is $L_\Phi$-Lipschitz continuous, that is, $|\nabla_\pi R(\pi_1 \cdot \Phi) - \nabla_\pi R(\pi_2 \cdot \Phi)| \leq L_\Phi\, |\pi_1 - \pi_2|,\ \forall \pi_1, \pi_2$.*

**Theorem 3.1.** *Presume Assumption 3.1. For $\delta \in \mathbb{R}_{++}$, consider the following set of parameters $\mathcal{F}_\delta$:*

$$\mathcal{F}_\delta \coloneqq \left\{ (\pi, \Phi) \ \middle|\ \sum_{e \in E_{\text{train}}} R_e(\pi \cdot \Phi) \leq \delta \right\}.$$

*Choose $\delta$ such that $\mathcal{F}_\delta \neq \varnothing$. Then,*

$$|\nabla_\pi R_e(\pi \cdot \Phi)|^2 \leq 2 L_\Phi \delta, \quad \forall (\pi, \Phi) \in \mathcal{F}_\delta, \forall e \in E_{\text{train}}.$$

*Proof.* See Appendix A. $\square$

Theorem 3.1 suggests that achieving sufficiently low training risk necessarily leads to a small gradient penalty term $|\nabla_\pi R_e(\pi \cdot \Phi)|^2$. However, the set $\mathcal{F}_\delta$ defined by this low-risk condition includes not only the desired invariant solutions but also "shotcut" solutions relying on spurious features. In other words, Theorem 3.1 indicates that these shortcut solutions can also achieve a small gradient penalty without restriction. Specifically, if such a $\Phi$, in conjunction with some $\pi$, results in a small ERM loss at the upper level of (5), then according to Theorem 3.1, the lower-level constraint in (5) will also be satisfied for some $\epsilon \ (\geq 2 L_\Phi \delta)$. As a

result, the spurious-feature extractor $\Phi$ may be incorrectly accepted as an invariant-feature one. The claim of Theorem 3.1 is justified by recent studies on over-parameterized settings where it has been shown that even when the condition $|\nabla_\pi R_e(\pi \cdot \Phi)|^2 = 0$ is satisfied, the learned feature extractor $\Phi$ may still depend on spurious features (Lin et al., 2022; Zhou et al., 2022).

Therefore, to ensure that IRM effectively identifies invariant features rather than spurious correlations, the ideal scenario is that $\mathcal{F}_\delta$ exclusively contains invariant solutions. This condition requires increasing the diversity of training environments, thereby invalidating shared spurious correlations and preventing shortcut solutions from simultaneously achieving low training risk.

## 4 Proposed Method

The preceding discussion reveals a fundamental vulnerability of IRMv1, which becomes pronounced in practical settings where achieving adequate diversity among training environments is inherently difficult. This section proposes a method to enhance the diversity of the training data *without* expanding set $E_{\text{train}}$ through data generation. Motivated by Section 3, two novel loss functions are introduced to address the aforementioned vulnerability of IRMv1.

### 4.1 Risk extrapolation

Section 3 suggests that increasing the diversity of $E_{\text{train}}$ is essential for the practical deployment of IRMv1. When $E_{\text{train}}$ exhibits limited diversity and the generation of synthetic data is not feasible, it becomes desirable to expand $E_{\text{train}}$ with pseudo-unseen environments and to perform optimization over this augmented set. Inspired by the work of (Krueger et al., 2021), the present study explores the generation of pseudo-unseen environments through algorithmic design, without the creation of additional data. As observed in (Krueger et al., 2021), risk $R_e(x, y)$ is linear with respect to the PDF $P_e(x, y)$, implying that an affine combination of risks across different distributions corresponds to the risk associated with a mixture of those distributions. Leveraging this insight, (Krueger et al., 2021) proposed a method of loss extrapolation through affine combinations—negative coefficients are allowed—of the individual environment losses $R_e(\cdot)$, thereby representing distributions outside the span of the original training environments. A subsequent min-max optimization of the risk over this extrapolated space enabled training with respect to pseudo-unseen environments.

### 4.2 Robustifying invariant learning through penalty extrapolation

The original IRMv1 regularization loss, $|\nabla_\pi R_e(\pi \cdot \Phi)|^2$, involves computing the gradient of the expected loss and taking its squared norm. Consequently, the loss is nonlinear with respect to the data distribution $P_e(x, y)$. This non-linearity presents challenges when attempting to extend the extrapolation approach of (Krueger et al., 2021), which relies on linear combinations of risks, to the IRMv1 setting.

To address this non-linearity issue, the following loss is introduced:

$$\mathcal{J}_{\text{IRM},e}(\pi, \Phi) \coloneqq \mathbb{E}_{(x,y) \sim P_e} \left\{ |\nabla_\pi \ell(\pi \cdot \Phi(x), y)|^2 \right\}, \quad \forall (\pi, \Phi), \forall e \in E_{\text{train}}. \tag{6}$$

In contrast to $|\nabla_\pi R_e(\pi \cdot \Phi)|^2$—recall that $R_e(\pi \cdot \Phi) = \mathbb{E}_{(x,y) \sim P_e} \left\{ \ell(\pi \cdot \Phi(x), y) \right\}$—notice that the squared norm of the gradient is computed prior to taking the expectation under $P_e(x, y)$ in (6). In this way, the loss in (6) is linear with respect to the PDF of the data, which in turn facilitates the implementation of the extrapolation ideas of (Krueger et al., 2021). The following shows that $\mathcal{J}_{\text{IRM},e}(\cdot, \cdot)$ "majorizes" $|\nabla_\pi R_e(\cdot, \cdot)|^2$.

**Lemma 4.1.** *For any risk function $R_e(\pi \cdot \Phi) = \mathbb{E}_{(x,y) \sim P_e} \left\{ \ell(\pi \cdot \Phi(x), y) \right\}$, the following holds true:*

$$|\nabla_\pi R_e(\pi \cdot \Phi)|^2 \leq \mathcal{J}_{\text{IRM},e}(\pi, \Phi), \quad \forall (\pi, \Phi), \forall e \in E_{\text{train}}.$$

*Proof.* See Appendix B. $\qquad\square$

For notational convenience, and motivated by the discussion in Section 2.5, let

$$\mathcal{J}_{\text{IRMv1},e}(\Phi) \coloneqq \mathcal{J}_{\text{IRM},e}(1.0, \Phi), \quad \forall \Phi, \forall e \in E_{\text{train}}.$$

Based on $\mathcal{J}_{\mathrm{IRMv1},e}$, this work proposes the following losses (7) and (9) instead of $|\nabla_\pi R_e(\pi \cdot \Phi)|^2$ in (IRMv1). First in order is the loss

$$\mathcal{C}_{\mathrm{mm}}(\Phi) := \max_{(\alpha_e)_{e \in E_{\mathrm{train}}} \in \mathcal{A}} \sum_{e \in E_{\mathrm{train}}} \alpha_e \, \mathcal{J}_{\mathrm{IRMv1},e}(\Phi)$$

$$= (1 - \alpha_{\min} |E_{\mathrm{train}}|) \max_{e \in E_{\mathrm{train}}} \mathcal{J}_{\mathrm{IRMv1},e}(\Phi) + \alpha_{\min} \sum_{e \in E_{\mathrm{train}}} \mathcal{J}_{\mathrm{IRMv1},e}(\Phi) , \qquad (7)$$

where $\boldsymbol{\alpha} := (\alpha_e)_{e \in E_{\mathrm{train}}}$ stands for a tuple of length equal to the cardinality $|E_{\mathrm{train}}|$ of $E_{\mathrm{train}}$,

$$\mathcal{A} := \left\{ \boldsymbol{\alpha} \in \mathbb{R}^{|E_{\mathrm{train}}|} \mid \sum_{e \in E_{\mathrm{train}}} \alpha_e = 1 , \, \alpha_{\min} \leq \alpha_e \right\} \qquad (8)$$

is the intersection of the affine set $\{ \boldsymbol{\alpha} \in \mathbb{R}^{|E_{\mathrm{train}}|} \mid \sum_{e \in E_{\mathrm{train}}} \alpha_e = 1 \}$ with the bound constraints $\{ \boldsymbol{\alpha} \in \mathbb{R}^{|E_{\mathrm{train}}|} \mid \alpha_{\min} \leq \alpha_e \}$, and the user-defined parameter $\alpha_{\min}$, which *may take negative values,* dictates the extent of extrapolation. The proof of the second equality in (7) is provided in Appendix C. The intuition behind (7) is to select, from a family of induced extrapolated pseudo-unseen environments, the distribution that maximizes the penalty. This strategy effectively simulates a broader set of training environments—without the need for synthetic or augmented data—thereby helping to mitigate overfitting to spurious features, even when the diversity of training environments is limited.

Furthermore, (Krueger et al., 2021) mentions that, in addition to extrapolation, simply adding the variance of risks across training environments as a regularization term proves to be stable and effective. To this end, the following alternative loss is also proposed:

$$\mathcal{C}_{\mathrm{v}}(\Phi) := \gamma \cdot \mathrm{Var}( \{ \mathcal{J}_{\mathrm{IRMv1},e}(\Phi) \mid e \in E_{\mathrm{train}} \} ) + \sum_{e \in E_{\mathrm{train}}} \mathcal{J}_{\mathrm{IRMv1},e}(\Phi) , \qquad (9)$$

where $\mathrm{Var}(S)$ represents the empirical variance over a finite set $S$ of real values, defined by

$$\mathrm{Var}(S) := \frac{1}{|S|} \sum_{s \in S} (s - \bar{s})^2 , \quad \bar{s} := \frac{1}{|S|} \sum_{s \in S} s ,$$

and $\gamma$ is a non-negative scalar that serves as a hyperparameter to determine the extent of regularization on the variance of the penalty.

To summarize, the following tasks, based on (7) and (9), are proposed as alternatives to the popular (IRMv1):

$$\min_{\Phi \colon \mathcal{X} \to \mathcal{Y}} \sum_{e \in E_{\mathrm{train}}} R_e(\Phi) + \lambda \, \mathcal{C}_{\mathrm{mm}}(\Phi) , \qquad (\text{mm-IRMv1})$$

$$\min_{\Phi \colon \mathcal{X} \to \mathcal{Y}} \sum_{e \in E_{\mathrm{train}}} R_e(\Phi) + \lambda \, \mathcal{C}_{\mathrm{v}}(\Phi) , \qquad (\text{v-IRMv1})$$

where $\lambda \in \mathbb{R}_{++}$ is a user-defined regularization parameter.

# 5 Numerical Tests

## 5.1 Structural equation models

Here, we conduct experiments using structural equation models (SEMs) in a scenario where the training environments are highly similar to each other, so that spurious features can misleadingly reduce the training loss.

### 5.1.1 Setting

The following SEM, taken from (Arjovsky et al., 2020), is considered:

$$\mathbf{x}_e^{\mathrm{inv}} \sim \mathcal{N}(\mathbf{0}_d, e^2 \mathbf{I}_d) ,$$
$$y_e := \mathbf{1}_d^\top \mathbf{x}_e^{\mathrm{inv}} + u , \quad u \sim \mathcal{N}(0, I_1) ,$$

$$\mathbf{x}_e^{\mathrm{spu}} \coloneqq y_e \cdot \mathbf{1}_d + \mathbf{v}_e, \quad \mathbf{v}_e \sim \mathcal{N}(\mathbf{0}_d, e^2 \mathbf{I}_d). \tag{10}$$

where $\mathbf{x}_e^{\mathrm{inv}}, \mathbf{x}_e^{\mathrm{spu}}$ and $y_e$ are $d \times 1$ vector-valued and scalar-valued realizations of random variables, respectively, $\mathcal{N}(\mathbf{0}, e^2 \mathbf{I}_d)$ stands for the $d$-dimensional normal PDF with mean the $d \times 1$ all-zero vector $\mathbf{0}_d$ and covariance matrix $e^2 \mathbf{I}_d$, $\mathbf{I}_d$ is the $d \times d$ identity matrix, $\mathbf{1}_d$ is the $d \times 1$ all-one vector, and $\top$ denotes vector/matrix transposition. Each environment is uniquely characterized by a distinct real value $e$.

The following estimation task is considered: given the $2d \times 1$ vector $\mathbf{x}_e \coloneqq [\mathbf{x}_e^{\mathrm{inv}\top}, \mathbf{x}_e^{\mathrm{spu}\top}]^\top$, estimate $y_e$ by $\hat{y}_e \coloneqq \hat{\mathbf{w}}_{\mathrm{inv}}^\top \mathbf{x}_e^{\mathrm{inv}} + \hat{\mathbf{w}}_{\mathrm{spu}}^\top \mathbf{x}_e^{\mathrm{spu}}$, where the $d \times 1$ parameter vectors $\hat{\mathbf{w}}_{\mathrm{inv}}, \hat{\mathbf{w}}_{\mathrm{spu}}$ of the estimation model need to be identified. In the case where $(\hat{\mathbf{w}}_{\mathrm{inv}}, \hat{\mathbf{w}}_{\mathrm{spu}}) = (\mathbf{1}_d, \mathbf{0}_d)$, that is, estimation is based solely on the invariant feature $\mathbf{x}_e^{\mathrm{inv}}$, then the estimation error $y_e - \hat{y}_e = y_e - \mathbf{1}_d^\top \mathbf{x}_e^{\mathrm{inv}} = u \sim \mathcal{N}(0, 1)$ becomes environment invariant, so that $\mathbb{E}\{y_e - \hat{y}_e\} = 0$ and $\mathbb{E}\{(y_e - \hat{y}_e)^2\} = 1$, regardless of the value of $e$. On the other hand, any attempt to utilize the spurious $\mathbf{x}_e^{\mathrm{spu}}$ in the estimation model by using a non-zero $\hat{\mathbf{w}}_{\mathrm{spu}}$ renders the estimation error environment dependent due to (10).

Tests are conducted for $d = 5$. To measure invariance, the causal error $(1/d)\|\hat{\mathbf{w}}_{\mathrm{inv}} - \mathbf{1}_d\|^2$ and the non-causal one $(1/d)\|\hat{\mathbf{w}}_{\mathrm{spu}} - \mathbf{0}_d\|^2$ are employed (Arjovsky et al., 2020). Set $E_{\mathrm{train}}$ comprises two environments, and three settings are investigated for $E_{\mathrm{train}}$: $\{0.2, 2\}, \{0.2, 1\}, \{0.2, 0.6\}$. As the two values of $e$ are getting closer to each other, $\mathbf{x}_e^{\mathrm{spu}}$ becomes spuriously invariant within $E_{\mathrm{train}}$, making it increasingly difficult to identify $\mathbf{x}_e^{\mathrm{inv}}$. The model was trained using the mean squared error (MSE) loss function, and hyperparameter tuning was performed based on validation data within the training environment. Further details are provided in Section D.1.

### 5.1.2 Results

Results are shown in Tables 1 and 6. A common trend observed across all methods is that as the $e$ values in the training environments become more similar, the error rates increase, making it more challenging to achieve invariance. However, while IRMv1 experiences particularly severe performance degradation, the proposed methods exhibit consistent improvements across all settings. Notably, mm-IRMv1 achieves the most substantial gains, with reductions of up to approximately 72% in causal error and 56% in non-causal error. Results under settings with a larger number of training environments but limited diversity are presented in Table 6. Even under these conditions, the proposed methods consistently outperform IRMv1. These findings confirm—from the perspective of learned parameters—that extrapolating the IRMv1 penalty fosters invariant learning, even when training environment diversity is limited.

Table 1: Invariance errors in SEMs. Even in scenarios where the training environments are similar—making it difficult to eliminate spurious features—the proposed methods, particularly mm-IRMv1, consistently achieve substantial improvements over the IRMv1 baseline. Percentages indicate performance changes relative to IRMv1.

| | $E_{\mathrm{train}} = \{0.2, 2\}$ | | $E_{\mathrm{train}} = \{0.2, 1\}$ | | $E_{\mathrm{train}} = \{0.2, 0.6\}$ | |
| --- | --- | --- | --- | --- | --- | --- |
| | causal err ($\downarrow$) | non-causal err ($\downarrow$) | causal err ($\downarrow$) | non-causal err ($\downarrow$) | causal err ($\downarrow$) | non-causal err ($\downarrow$) |
| IRMv1 | $0.487 \pm 0.840$ | $0.205 \pm 0.351$ | $0.798 \pm 0.152$ | $0.464 \pm 0.026$ | $1.418 \pm 0.091$ | $0.686 \pm 0.068$ |
| v-IRMv1 (Ours) | $\mathbf{0.414 \pm 0.650}$ (-15.0%) | $0.218 \pm 0.330$ (+6.3%) | $\mathbf{0.503 \pm 0.137}$ (-36.9%) | $\mathbf{0.360 \pm 0.082}$ (-22.4%) | $\mathbf{1.151 \pm 0.019}$ (-18.8%) | $\mathbf{0.581 \pm 0.027}$ (-15.3%) |
| mm-IRMv1 (Ours) | $\mathbf{0.218 \pm 0.373}$ (-55.2%) | $\mathbf{0.131 \pm 0.224}$ (-36.1%) | $\mathbf{0.222 \pm 0.059}$ (-72.2%) | $\mathbf{0.206 \pm 0.059}$ (-55.6%) | $\mathbf{1.006 \pm 0.311}$ (-29.1%) | $\mathbf{0.564 \pm 0.153}$ (-17.8%) |

## 5.2 Vision Datasets

The proposed framework is evaluated in the more typical and over-parameterized deep-learning scenarios for computer vision.

### 5.2.1 Setting

The proposed methods are evaluated on four vision classification datasets: Colored MNIST (CMNIST) (Arjovsky, 2020), Colored FashionMNIST (CFMNIST) (Ahuja et al., 2020), PACS (Li et al., 2017), VLCS (Torralba & Efros, 2011), ImageNet (Peng et al., 2019), and Camelyon17 (Bandi et al., 2018). Out-of-distribution (OOD) datasets generally differ in the level of generalization difficulty, depending on the nature of

Table 2: Test accuracy (%) is reported for the four datasets individually, along with their average. For each method, "base" refers to the original IRM variant, while "mm" and "v" denote the versions that incorporate the penalties defined in (7) and (9), respectively. On average, the proposed approach enhances performance across all IRM variants, with the "v" variant consistently outperforming the original. Percentages indicate performance changes relative to the baseline method.

| Method | | CMNIST | CFMNIST | PACS | VLCS | DomainNet | Camelyon17 | Avg. |
|--------|------|--------|---------|------|------|-----------|-----------|------|
| ERM | | 30.9±0.6 | 28.4±0.1 | 76.7±0.6 | 57.7±0.3 | 63.9±0.4 | 96.7±0.2 | 59.1 |
| IRMv1 | base | 64.7±0.5 | 74.3±1.1 | 75.5±1.4 | 58.4±0.5 | 63.9±0.6 | 95.8±0.1 | 72.1 |
| | v | 68.1±0.4 | 74.8±0.5 | 75.9±3.8 | 58.4±2.1 | 64.0±0.4 | 96.3±0.7 | **72.9** (+1.1%) |
| | mm | 66.8±0.6 | 73.5±0.4 | 72.7±3.9 | 59.0±1.1 | 64.0±0.4 | 96.0±0.8 | 72.0 (-0.1%) |
| BIRM | base | 66.5±0.5 | 75.5±0.9 | 76.3±2.2 | 56.7±1.9 | 64.0±0.8 | 96.0±0.6 | 72.5 |
| | v | 69.0±0.5 | 75.9±0.7 | 76.5±3.1 | 56.6±1.8 | 64.3±0.6 | 96.6±0.4 | **73.2** (+1.0%) |
| | mm | 66.2±0.6 | 73.4±0.2 | 73.4±0.2 | 56.7±0.6 | 64.7±0.3 | 96.1±0.4 | 71.8 (-1.0%) |
| BLO | base | 67.2±0.4 | 67.3±2.2 | 71.3±1.9 | 51.2±5.4 | 63.2±0.2 | 96.5±0.3 | 69.5 |
| | v | 67.0±1.9 | 70.5±2.1 | 71.4±4.6 | 53.8±5.4 | 63.3±0.1 | 96.3±0.1 | **70.4** (+1.3%) |
| | mm | 67.9±0.5 | 71.2±1.8 | 69.4±1.2 | 53.4±6.0 | 63.1±0.2 | 96.5±0.1 | **70.3** (+1.1%) |

Table 3: Test ECE is reported for the four datasets and their average. For each method, "base" denotes the original IRM variant, while "mm" and "v" refer to the versions incorporating the proposed penalties defined in (7) and (9), respectively. On average, the proposed approach improves calibration performance across all IRM variants. Percentages indicate performance changes relative to the baseline method.

| Method | | CMNIST | CFMNIST | PACS | VLCS | DomainNet | Camelyon17 | Avg. |
|--------|------|--------|---------|------|------|-----------|-----------|------|
| ERM | | 57.9±1.1 | 54.4±0.3 | 12.8±1.0 | 22.1±0.1 | 10.9±0.2 | 3.0±0.2 | 26.9 |
| IRMv1 | base | 10.4±0.3 | 17.8±0.6 | 14.7±1.5 | 22.8±0.2 | 10.8±0.2 | 3.7±0.7 | 13.4 |
| | v | 10.5±0.5 | 17.9±1.0 | 12.9±2.8 | 23.0±1.3 | 10.8±0.3 | 3.2±0.6 | **13.1** (-2.2%) |
| | mm | 10.5±0.9 | 13.8±0.6 | 16.6±3.6 | 19.8±0.6 | 11.1±0.2 | 3.4±0.5 | **12.5** (-6.7%) |
| BIRM | base | 10.1±0.9 | 19.6±0.7 | 12.9±1.3 | 22.8±1.0 | 11.0±0.1 | 3.2±0.3 | 13.3 |
| | v | 10.0±0.4 | 19.4±0.7 | 13.5±2.8 | 23.5±0.2 | 11.0±0.3 | 3.1±0.2 | 13.4 (+0.8%) |
| | mm | 9.5±0.4 | 13.4±0.6 | 13.4±0.6 | 22.9±1.5 | 11.1±0.0 | 3.2±0.4 | **12.3** (-7.5%) |
| BLO | base | 13.7±0.2 | 10.4±0.7 | 18.4±2.7 | 17.8±3.6 | 10.4±0.1 | 3.2±0.1 | 12.3 |
| | v | 11.6±1.7 | 11.6±1.7 | 18.6±6.0 | 15.6±1.9 | 10.6±0.1 | 3.4±0.3 | **11.9** (-3.3%) |
| | mm | 13.0±0.2 | 13.7±2.3 | 19.0±2.3 | 14.1±2.9 | 10.6±0.1 | 3.1±0.1 | 12.3 (±0.0%) |

the distributional shift. Specifically, CMNIST and CFMNIST exhibit correlation shifts, while the remaining datasets are characterized by diversity shifts (Ye et al., 2022). For the feature extractor $\Phi$, we used a pre-trained ResNet-50 (He et al., 2016) for ImageNet and Camelyon17, and a ResNet-18(He et al., 2016) trained from scratch for the remaining datasets.

The evaluation metrics include accuracy, expected calibration error (ECE), and adapted ECE (ACE) (Nixon et al., 2019), all computed on the test environments. While accuracy measures predictive performance, ECE and ACE specifically assess the calibration of the model's confidence—that is, how well the predicted confidence aligns with actual correctness. These calibration metrics provide a more nuanced understanding of model performance. Recent studies have shown that confidence calibration across multiple environments guarantees the optimality of IRM (Wald et al., 2021), and their empirical connections have been demonstrated by several works (Ovadia et al., 2019; Immer et al., 2021; Yoshida & Naganuma, 2024; Naganuma et al., 2025).

The competing methods include empirical risk minimization (ERM), IRMv1, and two recent state-of-the-art IRM variants: Bayesian IRM (BIRM) (Lin et al., 2022) and BLOC-IRM (BLO) (Zhang et al., 2023). Since BIRM and BLO are based on the losses $\mathcal{J}_{\mathrm{IRMv1},e}(\Phi)$ and $\mathcal{J}_{\mathrm{IRM},e}(\Phi)$, respectively, they are compatible with the proposed extrapolation method. Accordingly, in addition to IRMv1, the effectiveness of the extrapolation approach is also evaluated when integrated with BIRM and BLO.

Additional details can be found in Appendix D.

Table 4: Test adapted calibration error (ACE) is reported for the four datasets, along with their average. For each method, "base" denotes the original IRM variant, while "mm" and "v" represent the variants incorporating the proposed penalties defined in (7) and (9), respectively. On average, the proposed approach improves calibration performance across all IRM variants. Percentages indicate performance changes relative to the baseline method.

| Method | | CMNIST | CFMNIST | PACS | VLCS | DomainNet | Camelyon17 | Avg. |
|--------|------|---------|---------|------|------|-----------|------------|------|
| ERM | | $57.8\pm1.1$ | $54.5\pm0.3$ | $12.3\pm0.7$ | $22.0\pm0.1$ | $10.2\pm0.2$ | $2.5\pm0.2$ | 26.6 |
| IRMv1 | base | $14.0\pm0.3$ | $20.2\pm0.1$ | $14.1\pm1.7$ | $22.6\pm0.1$ | $10.1\pm0.2$ | $3.1\pm0.7$ | 14.0 |
| | v | $14.0\pm0.4$ | $20.2\pm0.3$ | $12.4\pm2.6$ | $22.6\pm1.4$ | $10.2\pm0.1$ | $2.6\pm0.5$ | **13.7** (-2.1%) |
| | mm | $13.8\pm1.0$ | $17.7\pm0.5$ | $16.1\pm3.7$ | $19.9\pm0.7$ | $10.4\pm0.1$ | $2.9\pm0.5$ | **13.5**(-3.6%) |
| BIRM | base | $13.8\pm1.0$ | $21.0\pm0.4$ | $12.4\pm1.1$ | $22.9\pm0.9$ | $10.2\pm0.1$ | $2.6\pm0.2$ | 13.8 |
| | v | $13.3\pm0.5$ | $20.7\pm0.5$ | $13.2\pm2.8$ | $23.1\pm0.4$ | $10.3\pm0.3$ | $2.5\pm0.2$ | 13.9 (+0.7%) |
| | mm | $13.0\pm0.0$ | $16.7\pm0.5$ | $12.4\pm1.1$ | $23.0\pm1.6$ | $10.3\pm0.1$ | $2.7\pm0.4$ | **13.0** (-5.8%) |
| BLO | base | $16.5\pm0.2$ | $14.9\pm0.3$ | $17.9\pm2.7$ | $19.1\pm3.5$ | $9.9\pm0.1$ | $2.7\pm0.1$ | 13.7 |
| | v | $14.6\pm1.3$ | $14.6\pm1.3$ | $18.0\pm6.2$ | $16.7\pm1.7$ | $10.0\pm0.2$ | $2.9\pm0.3$ | **12.8** (-6.6%) |
| | mm | $15.6\pm0.2$ | $17.2\pm2.2$ | $18.4\pm2.3$ | $15.6\pm2.6$ | $10.0\pm0.1$ | $2.6\pm0.0$ | **13.2** (-3.6%) |

### 5.2.2 Results

**Accuracy.** Table 7 presents the test accuracy of each method across the four datasets, along with their averages. On average, at least one of the proposed variants consistently outperforms the original method for every IRM variant. Notably, the "v" variant consistently surpasses the original methods. Compared to the best-performing original baseline (the base variant of BIRM at 72.5%), the proposed approach achieves an improvement of 1.0% with the v-BIRM variant.

**Calibration Metrics.** Next, we show the results for calibration metrics in Tables 3 and 4. Specifically, Table 3 reports results for ECE, and Table 4 for ACE. Across both metrics, our extrapolation methods consistently improve upon each original variant. The improvement in calibration metrics suggests that our distributional extrapolation approach effectively mitigates the overconfidence observed in recent neural networks (Minderer et al., 2021). However, unlike the comparison based on accuracy, we did not observe a consistent trend regarding the superiority of "v" or "mm". Notably, when focusing solely on IRMv1, the variant "v" consistently outperformed the original one, aligning with the trend observed in accuracy.

**Extrapolation Mitigates IRMv1 Overfitting.** Furthermore, the study examines whether the IRMv1 penalty term leads to overfitting to the training environment, as discussed in Section 3, and whether the proposed methods effectively mitigate this issue using the CMNIST dataset. Figure 1 presents scatter plots showing the relationships between the IRMv1 penalty values in the training environment and various evaluation metrics. In these plots, blue circles represent the original IRMv1, green squares denote v-IRMv1, and red triangles correspond to mm-IRMv1. Each point reflects the recorded values at each epoch during the final 50 epochs for each method.

Notably, the penalty values for the original IRMv1 approached zero over the final 50 epochs; however, its performance on test environment metrics was inferior to that of the proposed methods. This finding supports the vulnerability of IRMv1 discussed in Section 3. In contrast, while the penalty values for the proposed methods did not decline as sharply as those of IRMv1, they consistently achieved superior performance across all test metrics, indicating improved out-of-distribution generalization. These results demonstrate that the proposed distribution extrapolation approach effectively prevents overfitting to the penalty in the training environment, even in scenarios with limited environmental diversity such as CMNIST, thereby promoting the learning of truly invariant features.

Additional visualizations and scatter plots on other IRM variants can be found in Appendix E.2

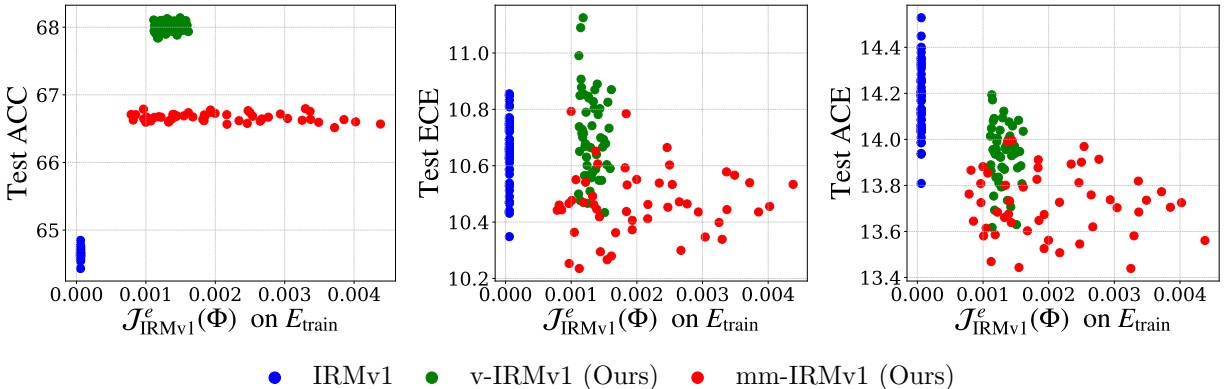

Figure 1: The figure illustrates the relationship between the IRMv1 penalty values in the training environment and the corresponding test evaluation metrics—accuracy, ECE, and ACE (from left to right)—on the CMNIST dataset. Each point represents the recorded values at each epoch during the final 50 epochs for each method. Although IRMv1 effectively reduces the training penalty to near zero, its performance across all test metrics remains suboptimal, reinforcing the vulnerability highlighted in Section 3. In contrast, the proposed distributional extrapolation methods mitigate overfitting of the IRMv1 penalty to the training environment and consistently yield improved performance across all evaluation metrics.

## 6 Related Work

### 6.1 Learning Methods for OOD Generalization

In recent practical applications of deep neural networks, violations of the i.i.d. (independent and identically distributed) assumption have become increasingly common, making the improvement of generalization performance under out-of-distribution (OOD) settings a critical challenge. To address this issue, learning algorithm-based approaches have emerged as one of the mainstream solutions.

Representation learning–based approaches are a widely studied area in the context of OOD generalization, aiming to extract invariant features from raw data that are independent of environmental variations. Prominent examples include IRM (Arjovsky et al., 2020), which relies on bi-level optimization, and domain-adversarial neural networks (DANN) (Ganin et al., 2016), which employ adversarial learning techniques.

In addition to representation learning, calibration-based approaches have also garnered increasing attention. Calibration refers to the alignment between a model's confidence and its predictive accuracy, and has been theoretically linked to OOD generalization (Wald et al., 2021; Yoshida & Naganuma, 2024). (Wald et al., 2021) demonstrated a theoretical connection between calibration and model invariance, showing that incorporating a calibration-focused regularization term into ERM can enhance OOD generalization performance.

Moreover, methods based on min-max optimization of the loss function have been proposed for OOD generalization. (Sagawa et al., 2020) applied the concept of distributionally robust optimization (DRO) (Ben-Tal et al., 2013)—which minimizes loss under the worst-case distribution within the training data—to deep learning and introduced group DRO. However, (Krueger et al., 2021) highlighted the overfitting tendency of traditional DRO approaches, which are confined to training environments, and proposed a method that performs DRO over a broader distribution set, closer to potential test environments, via loss extrapolation.

### 6.2 Variants of IRM

IRM (Arjovsky et al., 2020), introduced as a representation learning framework for OOD scenarios, formulates the objective as a bi-level optimization problem, which is known to be computationally challenging. To address this difficulty, a variety of approximation methods have been developed.

The most fundamental of these is IRMv1, which assumes a linear form for the predictor $\pi$ and reformulates the original problem as a single-level optimization via a penalization approach. However, IRMv1 has been

criticized for its tendency to overfit to training environments. In response, Bayesian IRM (BIRM) (Lin et al., 2022) adopts Bayesian inference to encourage invariance in the posterior distributions, aiming to mitigate overfitting. Information bottleneck-based IRM (IB-IRM) (Ahuja et al., 2022) improves the learning of invariant representations by applying the information bottleneck principle directly to the feature extractor $\Phi$ (Yoshida & Naganuma, 2024).

Alternative formulations based on game theory have also been proposed. IRM game (Ahuja et al., 2020) formulates the learning objective as a Nash equilibrium over environment-specific losses, while Pareto IRM (PAIR) (Chen et al., 2022) seeks Pareto-optimal trade-offs among the losses of ERM, IRMv1, and REx to balance in-distribution and OOD performance.

To address the scalability challenges posed by over-parameterized neural networks, (Zhou et al., 2022) demonstrated that IRMv1 can be effectively applied to large models by suppressing their fitting capacity through pruning, thereby preserving invariance. In addition, (Zhang et al., 2023) proposed a novel bi-level optimization strategy that simplifies the lower-level problem of IRM, diverging from conventional single-level approximation methods.

Despite their theoretical motivations, existing approximation methods have been reported to underperform compared to well-tuned ERM in practice (Gulrajani & Lopez-Paz, 2020), underscoring the need for more robust and principled IRM approximations. To this end, the present study introduces improved penalty formulations for IRMv1, aiming to enhance the performance of existing approximation methods that rely on the original IRMv1 penalty.

# 7 Discussion

## 7.1 v-IRMv1 or mm-IRMv1?

Although mm-IRMv1 consistently outperformed v-IRMv1 in the SEM-based experiments, the opposite trend was observed on realistic vision datasets, where v-IRMv1 demonstrated more stable and superior performance. This discrepancy is attributed to the numerical instability inherent in closed-form min-max optimization problems, as noted by (Krueger et al., 2021).

Specifically, the bottleneck arises from the max operator in (7): although each individual loss $\mathcal{J}_{\mathrm{IRMv1},e}(\Phi)$ may be smooth in $\Phi$ for all $e$, function $\max_{e \in E_{\mathrm{train}}} \mathcal{J}_{\mathrm{IRMv1},e}(\Phi)$ is not necessarily smooth and may exhibit a complex, non-differentiable landscape (graph) that is difficult to optimize. During training on realistic datasets, the environments $\arg\max_{e \in E_{\mathrm{train}}} \mathcal{J}_{\mathrm{IRMv1},e}(\Phi)$ that achieve the maximum value can change frequently and abruptly. This results into abrupt gradient shifts, numerical instability and increased difficulty in optimization.

The pronounced success of mm-IRMv1 in the SEM-based experiments is hypothesized to stem from the model's low dimensionality. While the vision-dataset experiments employed ResNet-18 in an overparameterized regime, the SEM setting used a model with only ten learnable parameters—five for $\hat{\mathbf{w}}_{\mathrm{inv}}$ and five for $\hat{\mathbf{w}}_{\mathrm{spu}}$—representing a substantially lower-dimensional hypothesis space. This significant reduction in degrees of freedom likely renders the model less sensitive to the non-smoothness introduced by the mm-IRMv1 objective, resulting in a less complex optimization landscape and more stable convergence.

As an ablation study on CMNIST, the proposed methods were evaluated using both a 3-layer MLP and ResNet-18. Table 5 illustrates how model dimensionality influences the test accuracy of mm-IRMv1. In the low-dimensional setting with the 3-layer MLP, mm-IRMv1 clearly outperforms v-IRMv1. However, in the higher-capacity, overparameterized ResNet-18, the effects of non-smoothness become more pronounced, and v-IRMv1 tends to yield superior performance.

# 8 Conclusions

This paper demonstrated that IRMv1, the most widely used approximation of invariant risk minimization (IRM), does not reliably achieve the intended invariance and is susceptible to overfitting due to reliance on spurious features. To address this limitation, this work proposed a novel approach that mitigates overfitting

Table 5: Test accuracy of mm-IRMv1 as a function of model dimensionality on CMNIST. In the low-dimensional and less flexible setting with a 3-layer MLP, mm-IRMv1 consistently outperforms v-IRMv1. However, as the model scales up and becomes increasingly overparameterized (e.g., ResNet-18), the non-smoothness introduced by mm-IRMv1 becomes more pronounced, allowing v-IRMv1 to achieve superior performance.

| Methods | Architecture (#params) | |
|---------|------------------------|---|
| | 3-layer MLP ($\approx$0.3M) | ResNet18 ($\approx$11.7 M) |
| IRMv1 | 65.7 $\pm$ 1.6 | 64.7 $\pm$ 0.5 |
| v-IRMv1 | 65.9 $\pm$ 1.6 | **68.1 $\pm$ 0.4** |
| mm-IRMv1 | **67.0 $\pm$ 1.0** | 66.8 $\pm$ 0.6 |

to limited training environments by extrapolating the IRMv1 penalty across data distributions, thereby pseudo-diversifying the training environments. The effectiveness of the proposed method was validated through both small-scale experiments using structural equation models (SEMs) and evaluations under recent overparameterized settings. This contribution offers a practical improvement to existing methods built upon the IRMv1 penalty and lays the groundwork for more principled approximations of IRM.

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

## Appendices

## A   Proof of Theorem 3.1

First, by the definition of $\mathcal{F}_\delta$,

$$\sum\nolimits_{e \in E_{\text{train}}} R_e(\pi \cdot \Phi) \le \delta\,, \quad \forall (\pi, \Phi) \in \mathcal{F}_\delta\,,$$

and the initial hypothesis that $R_e(\cdot) \ge 0$, it can be easily concluded that

$$R_e(\pi \cdot \Phi) \le \delta\,, \quad \forall (\pi, \Phi) \in \mathcal{F}_\delta\,, \forall e \in E_{\text{train}}\,.$$

Assume, for contradiction to the claim of the theorem, that there exist an $\hat{e} \in E_{\text{train}}$ and $(\hat{\pi}, \hat{\Phi}) \in \mathcal{F}_\delta$ such that

$$|\nabla_\pi R_{\hat{e}}(\hat{\pi} \cdot \hat{\Phi})| > \sqrt{2 L_{\hat{\Phi}} \delta}\,. \tag{11}$$

Hereafter, to simplify notation, $R_{\hat{e}}(\hat{\pi} \cdot \hat{\Phi})$ will be expressed as $R_{\hat{e}}(\hat{\pi})$ to stress the fact that it is a function of $\hat{\pi}$ since $\hat{\Phi}$ is fixed.

Let $\hat{\pi}' := \hat{\pi} - \eta \nabla_\pi R_{\hat{e}}(\hat{\pi} \cdot \hat{\Phi})$. By Assumption 3.1, the descent lemma (Nesterov, 2004, Lem. 1.2.3) suggests

$$
\begin{aligned}
R_{\hat{e}}(\hat{\pi}') &= R_{\hat{e}}\left( \hat{\pi} - \eta \nabla_\pi R_{\hat{e}}(\hat{\pi}) \right) \\
&\le R_{\hat{e}}(\hat{\pi}) + \nabla_\pi R_{\hat{e}}(\hat{\pi})^\top \left( -\eta \nabla_\pi R_{\hat{e}}(\hat{\pi}) \right) + \frac{L_{\hat{\Phi}}}{2} |-\eta \nabla_\pi R_{\hat{e}}(\hat{\pi})|^2 \\
&= R_{\hat{e}}(\hat{\pi}) - \eta |\nabla_\pi R_{\hat{e}}(\hat{\pi})|^2 + \frac{L_{\hat{\Phi}}}{2} \eta^2 |\nabla_\pi R_{\hat{e}}(\hat{\pi})|^2 \\
&= R_{\hat{e}}(\hat{\pi}) - (\eta - \frac{L_{\hat{\Phi}}}{2} \eta^2) |\nabla R_{\hat{e}}(\hat{\pi})|^2\,.
\end{aligned}
\tag{12}
$$

Let $\eta = 1/L_{\hat{\Phi}}$. Then

$$\eta - \frac{L_{\hat{\Phi}}}{2} \eta^2 = \frac{1}{L_{\hat{\Phi}}} - \frac{L_{\hat{\Phi}}}{2} \frac{1}{L_{\hat{\Phi}}^2} = \frac{1}{L_{\hat{\Phi}}} - \frac{1}{2 L_{\hat{\Phi}}} = \frac{1}{2 L_{\hat{\Phi}}}\,.$$

Hence, by (11) and (12),

$$
\begin{aligned}
R_{\hat{e}}(\hat{\pi}') &\le R_{\hat{e}}(\hat{\pi}) - \frac{1}{2 L_{\hat{\Phi}}} |\nabla_\pi R_{\hat{e}}(\hat{\pi})|^2 \\
&< \delta - \frac{1}{2 L_{\hat{\Phi}}} 2 L_{\hat{\Phi}} \delta \\
&= \delta - \delta = 0\,,
\end{aligned}
$$

which contradicts the initial hypothesis that $R_e(\cdot) \ge 0$. This completes the proof.

## B   Proof of Lemma 4.1

Notice that $\forall (\pi, \Phi)$,

$$
\begin{aligned}
|\nabla_\pi R_e(\pi \cdot \Phi)|^2 &= |\nabla_\pi \mathbb{E}_{(x,y) \sim P_e} \left\{ \ell(\pi \cdot \Phi(x), y) \right\}|^2 \\
&= |\mathbb{E}_{(x,y) \sim P_e} \left\{ \nabla_\pi \ell(\pi \cdot \Phi(x), y) \right\}|^2 \\
&\le \mathbb{E}_{(x,y) \sim P_e} \left\{ |\nabla_\pi \ell(\pi \cdot \Phi(x), y)|^2 \right\} \\
&= \mathcal{J}_{\text{IRM},e}(\pi, \Phi)\,,
\end{aligned}
$$

where the interchange between the expectation and the partial gradient in the second equality follows from the dominated convergence theorem, while the inequality is a direct application of Jensen's inequality (Ash & Doléans-Dade, 1999).

## C  Derivation of the closed-form expression of (7)

Problem (7) belongs to the class of linear-programming (LP) problems (Nocedal & Wright, 2006, (13.1)). By defining $\boldsymbol{\mu} := (\mu_e)_{e \in E_{\text{train}}} \in \mathbb{R}_+^{|E_{\text{train}}|}$ and for a $\varrho \in \mathbb{R}$, the Lagrangian function (Nocedal & Wright, 2006, (13.3)) for (7) becomes:

$$\mathcal{L}(\boldsymbol{\alpha}, \varrho, \boldsymbol{\mu}) = -\sum\nolimits_{e \in E_{\text{train}}} \alpha_e \, \mathcal{J}_{\text{IRMv1},e}(\Phi) + \varrho \left(1 - \sum\nolimits_{e \in E_{\text{train}}} \alpha_e\right) + \sum\nolimits_{e \in E_{\text{train}}} \mu_e(\alpha_{\min} - \alpha_e) \,,$$

where the minus sign is applied to $\mathcal{J}_{\text{IRMv1},e}(\Phi)$ to align with the convention that the Lagrangian function is typically formulated for minimization. It is well-known for LP problems that $\boldsymbol{\alpha}$ solves (7) if and only if there exist $(\varrho, \boldsymbol{\mu})$ such that the following Karush-Kuhn-Tucker (KKT) conditions are satisfied (Nocedal & Wright, 2006, (13.4)):

$$\text{(Stationarity:)} \qquad \frac{\partial \mathcal{L}}{\partial \alpha_e} = -\mathcal{J}_{\text{IRMv1},e}(\Phi) - \varrho - \mu_e = 0 \,, \quad \forall e \in E_{\text{train}} \,, \tag{13a}$$

$$\text{(Complementarity condition:)} \qquad \mu_e \left(\alpha_{\min} - \alpha_e\right) = 0 \,, \quad \forall e \in E_{\text{train}} \,, \tag{13b}$$

$$\text{(Primal feasibility:)} \qquad \sum\nolimits_{e \in E_{\text{train}}} \alpha_e = 1 \,, \quad \alpha_e \geq \alpha_{\min} \,, \tag{13c}$$

$$\text{(Dual feasibility:)} \qquad \mu_e \geq 0 \,, \quad \forall e \in E_{\text{train}} \,. \tag{13d}$$

Condition (13c) suggests $1 = \sum_e \alpha_e \geq \sum_e \alpha_{\min} = \alpha_{\min} |E_{\text{train}}| \Rightarrow \alpha_{\min} \leq 1/|E_{\text{train}}|$. Consequently, the following two cases are considered.

**Case 1:** $\alpha_{\min} = 1/|E_{\text{train}}|$. If there exists an $e \in E_{\text{train}}$ such that $\alpha_e > \alpha_{\min} = 1/|E_{\text{train}}|$, then (13c) yields $1 = \sum_e \alpha_e > \sum_e \alpha_{\min} = \sum_e 1/|E_{\text{train}}| = 1$, which is absurd. Hence, $\alpha_e = 1/|E_{\text{train}}|$, $\forall e \in E_{\text{train}}$. In such a case,

$$C_{\text{mm}}(\Phi) = \frac{1}{|E_{\text{train}}|} \sum\nolimits_{e \in E_{\text{train}}} \mathcal{J}_{\text{IRMv1},e}(\Phi) \,. \tag{14}$$

**Case 2:** $\alpha_{\min} < 1/|E_{\text{train}}|$. If $\alpha_e = \alpha_{\min}$, $\forall e \in E_{\text{train}}$, then the absurd result $1 = \sum_e \alpha_e = \sum_e \alpha_{\min} < \sum_e 1/|E_{\text{train}}| = 1$ is obtained, which suggests that there must exist at least one $e \in E_{\text{train}}$ such that $\alpha_e > \alpha_{\min}$.

  (i) Consider those $e \in E_{\text{train}}$ with $\alpha_e > \alpha_{\min}$—there certainly exists at least one such $e$. Then, (13b) yields $\mu_e = 0$, and (13a) leads to $\mathcal{J}_{\text{IRMv1},e}(\Phi) = -\varrho$.

  (ii) Consider those $e \in E_{\text{train}}$ with $\alpha_e = \alpha_{\min}$. Then, by (13a) and (13d), $\mu_e = -\mathcal{J}_{\text{IRMv1},e}(\Phi) - \varrho \geq 0 \Rightarrow \mathcal{J}_{\text{IRMv1},e}(\Phi) \leq -\varrho$.

The previous two points suggest $-\varrho = \max_{e \in E_{\text{train}}} \mathcal{J}_{\text{IRMv1},e}(\Phi)$. Upon defining $E_{\max} := \arg\max_{e \in E_{\text{train}}} \mathcal{J}_{\text{IRMv1},e}(\Phi)$ and its complement $E_{\max}^{\complement} := E_{\text{train}} \setminus E_{\max}$, it becomes also clear that if $e \in E_{\max}^{\complement}$, that is $\mathcal{J}_{\text{IRMv1},e}(\Phi) < -\varrho$, then necessarily $\alpha_e = \alpha_{\min}$. Because such an $\boldsymbol{\alpha}$ satisfies the KKT conditions, plugging it back into (7) yields

$$\begin{aligned}
C_{\text{mm}}(\Phi) &= \sum\nolimits_{e \in E_{\text{train}}} \alpha_e \, \mathcal{J}_{\text{IRMv1},e}(\Phi) \\
&= \sum\nolimits_{e \in E_{\max}} \alpha_e \, \max_{e \in E_{\text{train}}} \mathcal{J}_{\text{IRMv1},e}(\Phi) + \sum\nolimits_{e \in E_{\max}^{\complement}} \alpha_{\min} \, \mathcal{J}_{\text{IRMv1},e}(\Phi) \\
&= \left(1 - \sum\nolimits_{e \in E_{\max}^{\complement}} \alpha_{\min}\right) \max_{e \in E_{\text{train}}} \mathcal{J}_{\text{IRMv1},e}(\Phi) + \alpha_{\min} \sum\nolimits_{e \in E_{\max}^{\complement}} \mathcal{J}_{\text{IRMv1},e}(\Phi) \\
&= \left(1 - \alpha_{\min}|E_{\max}^{\complement}|\right) \max_{e \in E_{\text{train}}} \mathcal{J}_{\text{IRMv1},e}(\Phi) + \alpha_{\min} \sum\nolimits_{e \in E_{\text{train}}} \mathcal{J}_{\text{IRMv1},e}(\Phi) \\
&\quad - \alpha_{\min} \sum\nolimits_{e \in E_{\max}} \mathcal{J}_{\text{IRMv1},e}(\Phi) \\
&= \left(1 - \alpha_{\min}|E_{\max}^{\complement}|\right) \max_{e \in E_{\text{train}}} \mathcal{J}_{\text{IRMv1},e}(\Phi) + \alpha_{\min} \sum\nolimits_{e \in E_{\text{train}}} \mathcal{J}_{\text{IRMv1},e}(\Phi)
\end{aligned}$$

$$-\alpha_{\min}\,|E_{\max}|\,\max_{e\in E_{\text{train}}}\mathcal{J}_{\text{IRMv1},e}(\Phi)$$

$$= (1-\alpha_{\min}|E_{\text{train}}|)\max_{e\in E_{\text{train}}}\mathcal{J}_{\text{IRMv1},e}(\Phi)+\alpha_{\min}\sum_{e\in E_{\text{train}}}\mathcal{J}_{\text{IRMv1},e}(\Phi)\,,$$

where $|E_{\text{train}}|=|E_{\max}|+|E_{\max}^{\complement}|$ was used to deduce the last equality. Notice that this last expression for $C_{\text{mm}}$ covers also the case $\alpha_{\min}=1/|E_{\text{train}}|$ of (14), establishing thus (7).

## D  Implementation Details

### D.1  SEMs

All experiments are conducted on a single NVIDIA Tesla T4 GPU.

The experimental setup follows that of Arjovsky et al. (2020). Each environment contains 1,000 generated samples, and training is performed using full-batch updates. The number of training iterations is fixed at 20,000, with a constant learning rate of $1\times10^{-3}$. Penalty-related scaling factors are selected via grid search. The search ranges are: $1\times10^0, 1\times10^1$ for $\lambda$, $-1\times10^0, -5\times10^0, -1\times10^1$ for $\alpha_{\min}$, and $1\times10^0, 1\times10^1, 1\times10^2$ for $\gamma$. Results are averaged over three random seeds, and standard deviations are reported.

### D.2  Vision Datasets

All experiments are conducted using a single NVIDIA H100 GPU.

The datasets and their respective configurations are described as follows.

- **Colored MNIST (Colored FashionMNIST):** Colored MNIST (and Colored FashionMNIST) is a binary classification dataset derived from MNIST (or FashionMNIST), where digits 0–4 are assigned to class 0 and digits 5–9 to class 1. Each dataset consists of 70,000 samples (50,000 for training and 20,000 for testing), with input dimensions of $(1, 28, 28)$. The invariant features correspond to the original grayscale images, while the spurious features are encoded as a single-channel color (e.g., red or green) applied to each image. Typically, each class is associated with a specific color, but this color flips for a proportion $p$ of the samples. In the experiments, the training environments use $p = [10\%, 20\%]$, and the test environment uses $p = [90\%]$.

- **PACS:** PACS dataset contains 9,991 images of size $(3, 224, 224)$ across 7 classes and 4 distinct visual styles, each corresponding to a different domain. In the experiments, the `cartoon`, `photo`, and `sketch` domains are used for training, while the `art` domain is used for testing.

- **VLCS:** VLCS dataset consists of 10,729 images of size $(3, 224, 224)$ across 5 classes, drawn from four different sources. For the experimental setup, the `Caltech101`, `LabelMe`, and `VOC2007` domains are used for training, while `SUN09` serves as the test domain.

- **DomainNet:** DomainNet is a large-scale domain generalization benchmark with approximately 600,000 images of size $(3, 224, 224)$ across 345 classes. The images are sourced from 6 distinct domains. For our experiments, the `infograph`, `painting`, `quickdraw`, `real`, and `sketch` domains were used for the training environment, while the `clipart` domain was used for the test environment.

- **Camelyon17:** Camelyon17-WILDS dataset is a histopathology dataset focused on detecting tumor metastases in lymph node images. It contains approximately 450,000 tissue patches of size $(3,96,96)$ from 5 different hospitals. For our setup, `Hospital 0` was designated as the test domain, while the others were used as the training domains.

The Adam optimizer is used with a learning rate of $5\times10^{-4}$ across all datasets. For correlation shift experiments, models are trained using full-batch optimization for 500 epochs. For other settings, training is performed for 200 epochs on PACS and VLCS, and 10 epochs on others with a batch size of 32. Following

the protocol of Lin et al. (2022); Zhang et al. (2023), the penalty coefficient $\lambda$ is set to $1 \times 10^6$ for correlation shift experiments and $1 \times 10^0$ for diversity shift experiments.

The hyperparameters $\alpha_{\min}$ and $\gamma$ are tuned via grid search. The search ranges are:

- $\alpha_{\min} \in \left\{ -1 \times 10^{-1}, -2 \times 10^{-1}, \ldots, -1 \times 10^0 \right\}$

- $\gamma \in \left\{ 1 \times 10^{-1}, 2 \times 10^{-1}, \ldots, 1 \times 10^0 \right\}$

Model selection follows the "test-domain validation set" strategy (Gulrajani & Lopez-Paz, 2020) for correlation shift settings, and the "training-domain validation set" strategy (Gulrajani & Lopez-Paz, 2020) for diversity shift settings. All reported results are averaged over three random seeds, and standard deviations are also computed.

# E Additional Results

## E.1 Results on SEMs

Table 6: Invariance errors in SEMs. Even in scenarios where the training environments are similar—making it difficult to eliminate spurious features—the proposed methods, particularly mm-IRMv1, consistently achieve substantial improvements over the IRMv1 baseline.

| | $E_{\text{train}} = \{0.1, 0.5, 1\}$ | | $E_{\text{train}} = \{0.1, 0.4, 0.7, 1\}$ | | $E_{\text{train}} = \{0.1, 0.2, 0.3, 0.4, 0.5\}$ | |
| | causal err ($\downarrow$) | non-causal err ($\downarrow$) | causal err ($\downarrow$) | non-causal err ($\downarrow$) | causal err ($\downarrow$) | non-causal err ($\downarrow$) |
|---|---|---|---|---|---|---|
| IRMv1 | $0.679 \pm 0.093$ | $0.414 \pm 0.046$ | $0.986 \pm 0.513$ | $0.524 \pm 0.192$ | $1.179 \pm 0.424$ | $0.570 \pm 0.231$ |
| v-IRMv1 (Ours) | $\mathbf{0.597 \pm 0.024}$ | $\mathbf{0.381 \pm 0.014}$ | $\mathbf{0.689 \pm 0.320}$ | $\mathbf{0.402 \pm 0.094}$ | $\mathbf{0.776 \pm 0.567}$ | $\mathbf{0.330 \pm 0.195}$ |
| mm-IRMv1 (Ours) | $\mathbf{0.427 \pm 0.220}$ | $\mathbf{0.305 \pm 0.146}$ | $\mathbf{0.527 \pm 0.252}$ | $\mathbf{0.273 \pm 0.127}$ | $\mathbf{0.736 \pm 0.537}$ | $\mathbf{0.373 \pm 0.204}$ |

In Table 1, the number of training environments is limited to two. Additional experiments are conducted under alternative settings, with results presented in Table 6. Specifically, scenarios with three, four, and five training environments are evaluated. Across all cases, our methods consistently achieves substantial improvements.

## E.2 Relationship between IRM penalty and Test Performance

This section provides additional figures that illustrate the effectiveness of the proposed methods in mitigating overfitting of the IRM penalty, consistent with the results shown in Figure 1.

### E.2.1 BIRM

Figure 2 illustrates the impact of the proposed extrapolation method on BIRM for CMNIST. While all methods perform similarly in minimizing $\mathcal{J}_{\text{IRMv1},e}(\Phi)$ within the training environments, notable differences emerge in test performance, demonstrating that the proposed approach effectively mitigates overfitting to the training data. Furthermore, although the test performance of the original BIRM remains nearly constant, it exhibits instability in $\mathcal{J}_{\text{IRMv1},e}(\Phi)$ across the training environments.

### E.2.2 BLO

Figure 3 illustrates the impact of the proposed extrapolation method on BLO for CMNIST. While all methods achieve comparable values of $\mathcal{J}_{\text{IRMv1},e}(\Phi)$ in the training environments, substantial differences arise in test performance, indicating that the proposed approach effectively mitigates overfitting to the training data. Notably, both proposed methods improve calibration metrics relative to the original BLO, demonstrating that the distributional extrapolation technique reduces overconfidence.

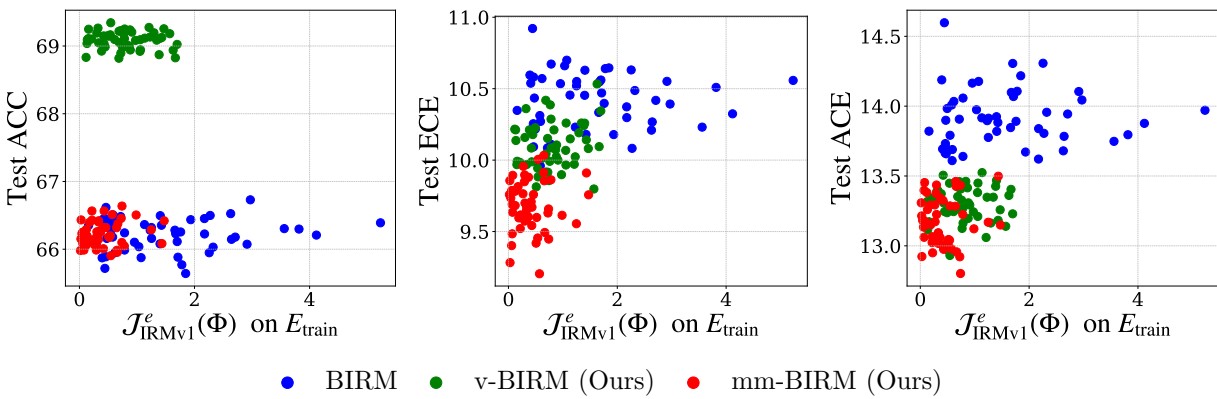

Figure 2: The relationship between the IRMv1 penalty values in the training environment and the corresponding test evaluation metrics (from left to right: accuracy, ECE, and ACE) on CMNIST. Each point represents the values recorded at each epoch during the last 50 epochs for each method.

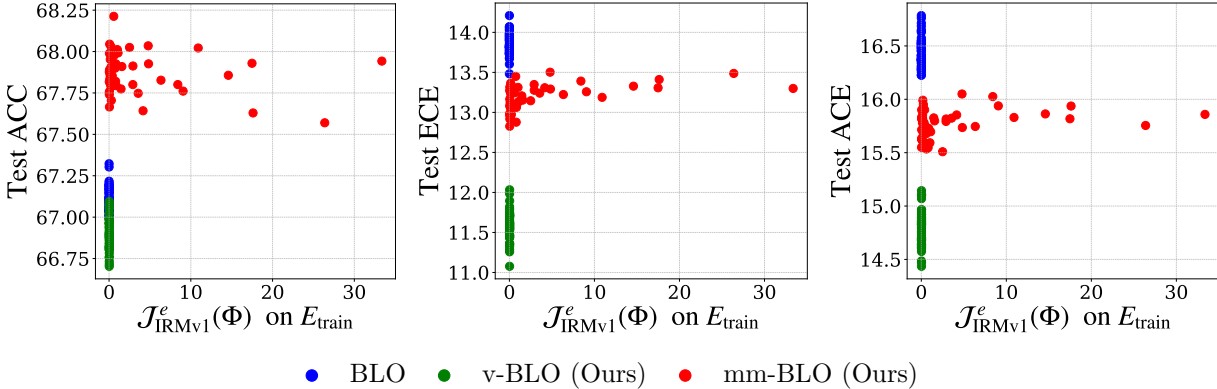

Figure 3: The relationship between the IRM penalty values in the training environment and the corresponding test evaluation metrics (from left to right: accuracy, ECE, and ACE) on CMNIST. Each point represents the values recorded at each epoch during the last 50 epochs for each method.

### E.2.3 Extrapolated IRMv1 and other existing methods

Figure 4 presents a comparison of v-IRMv1, mm-IRMv1, and other existing methods on CMNIST. Notably, v-IRMv1 maintains a consistently low $\mathcal{J}_{\text{IRMv1},e}(\Phi)$ in the training environments while achieving superior average test accuracy and calibration metrics compared to the other methods. This suggests that invariant learning is effectively achieved. In contrast, although IRMv1 reports relatively low calibration errors, it exhibits the lowest test accuracy overall. Additionally, both BIRM and BLO demonstrate suboptimal performance on at least one evaluation metric.

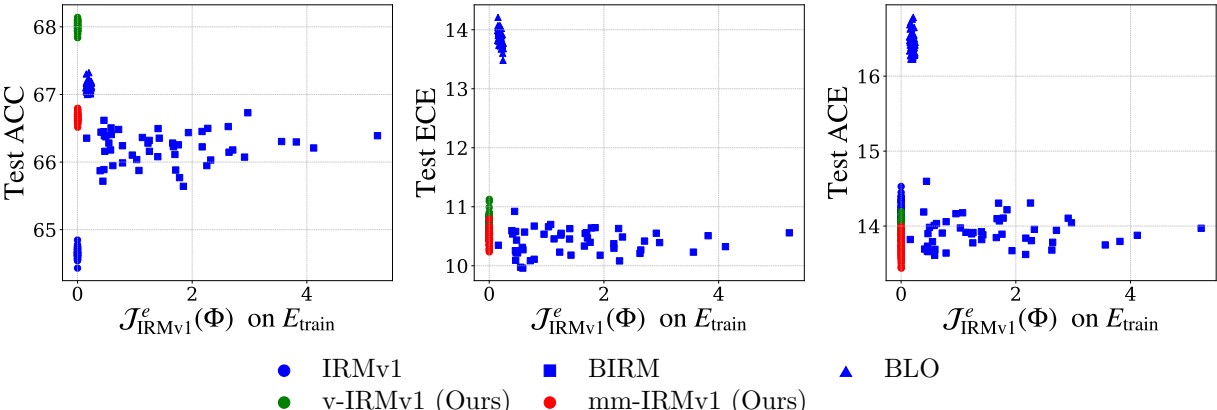

Figure 4: The relationship between the IRMv1 penalty values in the training environment and the corresponding test evaluation metrics (from left to right: accuracy, ECE, and ACE) on CMNIST. Each point represents the values recorded at each epoch during the last 50 epochs for each method.

Table 7: Average test ACC, ECE, and ACE under Correlation (Corr.) and Diversity (Div.) shifts. Corr. Shift includes CMNIST and CFMNIST; Div. Shift includes PACS, VLCS, DomainNet, and Camelyon17.

| Method | | Corr. Shift | | | Div. Shift | | |
|--------|------|------|------|------|------|------|------|
| | | ACC | ECE | ACE | ACC | ECE | ACE |
| ERM | | 29.7 | 56.2 | 56.2 | 73.8 | 12.2 | 11.8 |
| IRMv1 | base | 69.5 | 14.1 | 17.1 | 73.4 | 13.0 | 12.5 |
| | v | **71.5** | 14.2 | 17.1 | **73.7** | **12.5** | **12.0** |
| | mm | **70.2** | **12.2** | **15.8** | 72.9 | **12.7** | **12.3** |
| BIRM | base | 71.0 | 14.9 | 17.4 | 73.3 | 12.5 | 12.0 |
| | v | **72.5** | **14.7** | **17.0** | **73.5** | 12.8 | 12.3 |
| | mm | 69.8 | **11.5** | **14.9** | 72.7 | 12.7 | 12.1 |
| BLO | base | 67.3 | 12.1 | 15.7 | 70.6 | 12.5 | 12.4 |
| | v | **68.8** | **11.6** | **14.6** | **71.2** | **12.1** | **11.9** |
| | mm | **69.6** | 13.4 | 16.4 | 70.6 | **11.7** | **11.7** |

