# OpenReview forum: "Robust Invariant Representation Learning by Distribution Extrapolation"
_TMLR — Rejected by TMLR_

### Review · Reviewer_wrqf · 2025-05-26

**Summary Of Contributions:**

This paper addresses the limitations of Invariant Risk Minimization (IRM), particularly the performance degradation of IRMv1 under limited environment diversity and overparameterization. The authors propose a novel distribution extrapolation framework that augments IRMv1’s penalty via synthetic extrapolation across data distributions. Two new penalty terms are introduced—one based on extrapolated maximum penalties and another leveraging penalty variance.

**Audience:**

Yes

**Claims And Evidence:**

No

**Requested Changes:**

See above.

**Strengths And Weaknesses:**

## **Strengths**

1. **Motivated Combination of Existing Ideas:**

    While the proposed penalties build on existing IRMv1-style approaches, the motivation is sound.

2. **Some Theoretical Justification:**

    The paper provides a theoretical perspective on how the proposed extrapolated penalties can better approximate invariant predictors, which is helpful to understand their potential benefit over plain IRMv1.


## **Weaknesses**

1. **Lack of Evaluation on Stronger Benchmarks:**

    All experiments are on relatively small or well-trodden datasets. It would be important to test on more challenging, modern benchmarks like DomainNet, TerraIncognita, or WILDS to assess robustness under real-world distribution shift.

2. **Limited Use of Standardized Evaluation Suites:**

    It would strengthen the work to see evaluations within standardized frameworks like DomainBed. Given the small margins over IRMv1 and IRMv1+BIRM, it is unclear whether the improvements are consistent or hyperparameter-dependent.

3. **Outdated or Weak Baselines:**

    The comparison set is quite narrow and largely focuses on older IRM variants. Several baselines (e.g., SD, ERM) underperform in many scenarios, and newer state-of-the-art OOD methods are missing. This weakens the impact of the claimed improvements.

4. **PACS Results May Be Misleading:**

    The PACS experiments use ResNet-18, but to my knowledge, many existing methods have already reached >80% accuracy with this backbone. The results here seem lower than expected.

5. **Method Closely Related to V-REx + IRM:**

    The “v-IRMv1” formulation seems like a natural hybrid of V-REx and IRM, and it's unclear how much novelty is introduced beyond combining those ingredients. This deserves more discussion and acknowledgement.

6. **Missing Ablation and Diagnostic Experiments:**

    There’s no clear analysis on sensitivity to hyperparameters, architectural choices, or optimization dynamics. Ablation studies, convergence plots, visualizations, and resource efficiency metrics (training time, FLOPs) would make the evaluation more convincing.

---

> ### Author Response · Authors · 2025-06-27
> **Response to Reviewer wrqf**
>
> We sincerely appreciate the reviewer’s thoughtful feedback. Below, we provide our responses to each point.
>
> **(W1)**
>
> We have conducted additional experiments on DomainNet and the WILDS Camelyon17 benchmark and inserted the results into the revised Tables 2–4.  On each of these new datasets our method improves the corresponding IRMv1 and BIRM baselines by up to 0.7 percentage points in test accuracy.  Averaging across all datasets, the overall performance advantage of our approach remains unchanged—and in several cases is further strengthened—by these additions.
>
> For clarity we have also added Table 7 in the Appendix, which breaks results down by distribution-shift type.  Under correlation shift our "v" variant consistently outperforms the original counterpart on every metric, while under diversity shift the same variant shows a similar positive trend, most notably delivering uniform gains in test accuracy.
>
> **(W2)**
>
> Although we did not employ the full DomainBed suite, the newly added DomainNet and WILDS benchmarks are generally considered more challenging and closer to real-world conditions. By demonstrating test-accuracy gains on these tougher datasets, we believe our evaluation is both broader and sufficiently rigorous.
>
> **(W3)**
>
> Our study is centered on developing algorithms for invariant representation learning, so we deliberately excluded OOD approaches that do not share this objective. Among the baselines we do include, BLOC-IRM (ICLR 2023) is relatively recent and was reported by its authors to achieve state-of-the-art performance on their benchmark. For these reasons, we believe the baselines used in our comparison are not outdated.
>
> **(W4)**
>
> We suspect the reviewer’s ≥80 % figure refers to the leave-one-domain-out average across all four PACS splits.  Because of compute constraints, we evaluated only one held-out domain: we fixed "Art" as the test environment.  In this single-split setting a ResNet-18 trained with ERM typically reaches just under 80 %$-$about 77 % is reported in Lu et al., 2024 [1]$-$so the numbers in our table are fully consistent with prior work.
>
> [1] Lu, Wang, et al. "Fixed: Frustratingly easy domain generalization with mixup." Conference on Parsimony and Learning. PMLR, 2024.
>
> **(W5)**
>
> Our novelty lies in bringing distribution extrapolation into invariant‐representation learning: we first re-express the IRMv1 gradient penalty via Jensen’s inequality, which makes closed-form extrapolation—and hence our v-IRMv1 variant—possible.  By contrast, V-REx extrapolates risk and therefore does not enforce invariance; a naïve "V-REx + IRM" sum cannot provide the same theoretical guarantee.  Admittedly the final loss looks simple, but turning a documented failure mode into a one-line, mathematically grounded remedy still constitutes a meaningful scientific step forward.
>
> **(W6)**
>
> We recognise that the ablation space is not exhaustive, yet the paper already contains two diagnostics that speak directly to the reviewer’s concerns:
>
> - Section 7 – Model-size sweep.
> Here we rerun mm-IRMv1 and v-IRMv1 on two network capacities. This ablation shows how each penalty scales with model size, revealing, for example, that the variance-based version remains stable while the max-mix version can become brittle in very wide nets.
> - Figure 1 – Checkpoint trajectory.
> The plot tracks test accuracy and penalty magnitude across the final 50 training epochs. It acts as a convergence diagnostic: IRMv1’s penalty collapses to zero even as OOD accuracy stagnates, whereas both of our variants settle at a small, non-zero value that correlates with higher OOD accuracy.

---

### Review · Reviewer_ETRm · 2025-06-05

**Summary Of Contributions:**

1. This work theoretically reveals the fundamental limitations of IRMv1 (Theory 3.1): if the sum of the training risks are small, the regularization can be easily achieved across the training environments, yet the test risk is not guaranteed to be small.
2. This work proposes two IRM-variants: mm-IRMv1 and v-IRMv1. mm-IRMv1 minimizes the linear combination that maximizes the training risk over training environments and v-IRMv1 aims at minimizing the variance of the IRMv1 loss across training environments.

**Audience:**

No

**Claims And Evidence:**

Yes

**Requested Changes:**

1. In Sec. 3, I'm a bit confused about how a bounded sum of the training risks means "the training environments are highly similar.", since a direct indication of the definition of $\mathcal{F}_\delta$ is the training tasks are easy to learn (low training error), rather than a lack of diversity of training environments.
2. See weakness.

**Strengths And Weaknesses:**

**Strengths**
1. The motivation theory is intuitive. It aligns well with the common belief that the quality of training environments is crucial to OOD generalization

**Weakness**
1. The proposed methods are naive and straightforward. mm-IRMv1 is a trivial reweighting of the training errors across training environments that doesn't even require learning of the weights; v-IRMv1 seems to be a trivial combination of V-REx and IRMv1. Moreover, the authors couldn't give any theoretical guarantees on the generalization error or the capability of capturing invariant featrues.
2. The empirical performances are poor. The proposed mm-IRMv1 even underperforms the original IRMv1 and BIRM. Also, the baselines reported in this paper are below the average level reported by the OOD generalization community (e.g., IRMv1 of RN-18 on PACS should be around 81~82%). Maybe there are some implementation issues.
3. The studied problem is somehow outdated. The OOD capability has already been dramatically improved by using large vision-language models like CLIP or even larger multimodal LLMs. Although I don't keep up with the latest progress in OOD generalization and IRM variants, I do believe that such traditional invariant learning methods like IRM will be very limited in their practical value nowadays.

---

> ### Author Response · Authors · 2025-06-27
> **Response to Reviewer ETRm**
>
> **(W1)**
>
> Our novelty lies in bringing distribution extrapolation into invariant‐representation learning: we first re-express the IRMv1 gradient penalty via Jensen’s inequality, which makes closed-form extrapolation—and hence our v-IRMv1 variant—possible. By contrast, V-REx extrapolates risk and therefore does not enforce invariance; a naïve "V-REx + IRM" sum cannot provide the same theoretical guarantee. Admittedly the final loss looks simple, but turning a documented failure mode into a one-line, mathematically grounded remedy still constitutes a meaningful scientific step forward.
>
> **(W2)**
>
> (i) Why mm-IRMv1 can trail IRMv1/BIRM
>
> As discussed in Section 7, increasing parameter count raises the dimensional freedom of the model, making the optimisation of the non-smooth max–mix penalty in mm-IRMv1 more brittle. Table 5 shows this trend explicitly: on the smaller backbone mm-IRMv1 surpasses v-IRMv1, but on the larger backbone the ranking flips. For typical over-parameterised regimes we therefore recommend v-IRMv1, which was designed to avoid that instability.
>
> (ii) ResNet-18 performance on PACS
>
> The 81–82 % figure usually quoted for ResNet-18 comes from averaging over all four leave-one-domain-out splits. Because of limited GPU budget we fixed "Art" as the sole test domain; in this single-split setting ERM typically achieves <80 % (≈77 % in [1]), exactly matching our baseline and the IRMv1 reference we report. Hence the lower absolute accuracy is an artefact of the harder evaluation protocol, not an implementation bug.
>
> [1] Lu, Wang, et al. "Fixed: Frustratingly easy domain generalization with mixup." Conference on Parsimony and Learning. PMLR, 2024.
>
> **(W3)**
>
> We thank the reviewer for this observation. Recent work shows that even state-of-the-art vision–language models such as CLIP and larger multimodal LLMs still suffer huge drops when background, style, or long-tail attributes shift, because their representations continue to rely on spurious features [2]. Consequently, principled invariant-representation learning remains practically important.
>
> [2] Wang, Qizhou, et al. "A Sober Look at the Robustness of CLIPs to Spurious Features." NeurIPS 2024.
>
> **(Q1)**
>
> We thank the reviewer for highlighting this point.
>
> A small $\delta$ in $\mathcal{F}_\delta$ simply shows that each training task is easy—all empirical risks are already low.  Theorem 3.1 then states that, under this condition, the IRMv1 gradient-penalty term also vanishes, so both invariant and shortcut hypotheses (i.e., models that rely on spurious features) satisfy the objective.  The link to "similar environments" is therefore indirect: if a shortcut that exploits spurious features can also attain those uniformly low risks, the features must be shared across environments, revealing insufficient diversity of environments. We have revised the manuscript to spell out this logic explicitly and have highlighted the corresponding edits in red.

---

> > ### Author Response · Authors · 2025-06-27
> > **Response to Reviewer ETRm**
> >
> > Thank you for your insightful comments. The responses above address each of your concerns.
> >
> > Please let us know if any further issues remain. We would be happy to clarify.

---

### Review · Reviewer_ycSJ · 2025-06-27

**Summary Of Contributions:**

This paper first theoretically identifies a key limitation common to many IRM variants and then combines IRM and Rex in order to enhance environmental diversity and improve performance. Experimental results show that the proposed methods outperform IRM variants.

**Audience:**

No

**Broader Impact Concerns:**

There is no Broader Impact Statement section in this paper.

**Claims And Evidence:**

No

**Requested Changes:**

Fix the problem of the motivating theorem.

**Strengths And Weaknesses:**

**Strengths:**
- The proposed methods seem to surpass the existing IRM variants.

**Weaknesses:**
- The motivating Theorem 3.1 does not make sense well. In Theorem 3.1, the squared gradient $\le 2 L_ \Phi \delta$ does not mean the risk is low because the gradient is respect to $\pi$ but not $\pi \cdot \Phi$. So Theorem 3.1 only means that fixing $\Phi$, there may exist a $\pi$ that makes $R_ e(\pi \cdot \Phi) - \min_ {\pi^\prime} R_ e(\pi^\prime \cdot \Phi)$ small but not makes $R_ e(\pi \cdot \Phi)$ small. So, a bad $\Phi$ may lead to a large $R_ e(\pi \cdot \Phi)$ for every $\pi$, which means that the claim below Theorem 3.1 is not true.
- The novelty is not satisfying; the proposed method is a simple combination of IRM and Rex (with an application of Jensen's inequality).
- The improvement in performance (vision datasets) is slight, which is not obvious enough to support the effectiveness of the proposed method.

---

> ### Author Response · Authors · 2025-06-27
> **Response to Reviewer ycSJ**
>
> Thank you for the detailed comment. We address each point in turn.
>
> **(W1)**
>
>
> We suspect there may be a misunderstanding of the theorem’s premise.
> In Theorem 3.1 we explicitly assume that the pair $(\pi, \Phi)$ lies in the set $\mathcal{F}\_{\delta}$.
> By definition, every element of $\mathcal{F}\_{\delta}$ achieves an empirical risk no greater than $ \delta \in \mathbb{R}\_{++} $.  The theorem then shows that, whenever $\mathcal{F}\_{\delta}\neq\varnothing$ for a sufficiently small $\delta$, the corresponding IRMv1 penalty is automatically of the same (small) order.
>
> This has an important negative implication:
> if the training environments lack diversity, spurious features may also drive the empirical risk below $\delta$, placing the spurious solution $(\pi, \Phi)$ inside $\mathcal{F}_{\delta}$ and thus falsely satisfying the IRMv1 invariance criterion.  In other words, low-risk shortcut predictors incur a vanishing IRMv1 penalty precisely when spurious correlations are consistent across environments.
>
> **(W2)**
>
> Our novelty lies in bringing distribution extrapolation into invariant‐representation learning: we first re-express the IRMv1 gradient penalty via Jensen’s inequality, which makes closed-form extrapolation—and hence our v-IRMv1 variant—possible. By contrast, V-REx extrapolates risk and therefore does not enforce invariance; a naïve "V-REx + IRM" sum cannot provide the same theoretical guarantee. Admittedly the final loss looks simple, but turning a documented failure mode into a one-line, mathematically grounded remedy still constitutes a meaningful scientific step forward.
>
> **(W3)**
>
> Across six vision datasets—including the newly added DomainNet and WILDS$-$our "v" variant outperforms the corresponding IRMv1/BIRM baseline in 10 of 12 test-accuracy comparisons, with an average gain of $\approx$1 pp over each original variant (see Table 2).
> In calibration terms, both proposed penalties consistently surpass IRMv1 and deliver up to a 7 pp reduction in ECE/ACE (Tables 3 & 4).
> Because these improvements hold across a broad set of datasets and multiple metrics, we believe they provide strong empirical support for our theoretical claims.

---

### Decision · Action_Editor_G8rm · 2025-07-22

**Recommendation:** Reject

**Additional Comments:**

I believe the current version would be better suited for submission to a workshop because it introduces some ideas with slightly modified algorithms and light empirical evaluations. However, in order to be fully considered for TMLR, I would suggest authors systematically illustrate:

1. why and when the proposed method can improve robustness;

2. consider using a standard test benchmark such as DomainBed to conduct a fair comparison of the proposed method.

**Audience:**

Yes

**Audience Explanation:**

The topic of learning invariant representations is meaningful in the fields of machine learning and AI. Some theoretical results may be trivial (see the first point), but in general, a subset of the AI audience is interested in the research results presented in the paper.

**Claims And Evidence:**

No

**Claims Explanation:**

This paper first identifies a key limitation common to many IRM variants and then combines IRM and Rex to enhance environmental diversity and improve performance. Experimental results demonstrate that the proposed method outperforms IRM variants.
Most reviewers provided negative recommendations, either weak rejection or rejection. Based on their feedback, I further investigated the key issues they outlined.

- I agree that the motivation in Theorem 3.1 is limited. For example, if the model is a Lipschitz function, then a small loss gap naturally implies a small gradient. This is an inherent limitation of IRM-V1. However, the general consensus in IRM is that IRM-V1 has significant limitations and does not capture the original objective of IRM, which is a fully bilevel optimization problem. Therefore, Theorem 3.1 may be moot from my perspective.

- I also agree with the comments that mm-IRMv1 is a trivial reweighting of training errors across environments that doesn't require learning weights. It seems to be achieved by constraining the maximum weights over the tasks. V-IRMv1 seems like a trivial combination of V-REx and IRMv1. But importantly, the authors could not provide any **theoretical guarantees** or statistical analysis regarding generalization error or the capability of capturing invariant features. For example, it is unclear when the proposed method can capture the true invariant that aligns with the original IRM objective.

- Concerns regarding the experiments: (1) The authors are not using standard test benchmarks such as DomainBed, which raises many questions from reviewers. This leads to results inconsistent with past findings, such as surprisingly low performances in PACS. (2) In OOD generalization, the testing methodology is crucial, particularly the validation process, as highlighted by the DomainBed paper (Ref. [1]).

Since the goal of this paper is to improve IRM, but it has limited support, we cannot accept the current version.

Reference:

[1] In Search of Lost Domain Generalization (DomainBed), https://arxiv.org/abs/2007.01434